# Application and Prospect of Artificial Intelligence Methods in Signal Integrity Prediction and Optimization of Microsystems

**DOI:** 10.3390/mi14020344

**Published:** 2023-01-29

**Authors:** Guangbao Shan, Guoliang Li, Yuxuan Wang, Chaoyang Xing, Yanwen Zheng, Yintang Yang

**Affiliations:** 1School of Microelectronics, Xidian University, Xi’an 710071, China; 2Beijing Institute of Aerospace Control Devices, Beijing 100039, China

**Keywords:** microsystem, signal integrity, neural network, optimization algorithms

## Abstract

Microsystems are widely used in 5G, the Internet of Things, smart electronic devices and other fields, and signal integrity (SI) determines their performance. Establishing accurate and fast predictive models and intelligent optimization models for SI in microsystems is extremely essential. Recently, neural networks (NNs) and heuristic optimization algorithms have been widely used to predict the SI performance of microsystems. This paper systematically summarizes the neural network methods applied in the prediction of microsystem SI performance, including artificial neural network (ANN), deep neural network (DNN), recurrent neural network (RNN), convolutional neural network (CNN), etc., as well as intelligent algorithms applied in the optimization of microsystem SI, including genetic algorithm (GA), differential evolution (DE), deep partition tree Bayesian optimization (DPTBO), two stage Bayesian optimization (TSBO), etc., and compares and discusses the characteristics and application fields of the current applied methods. The future development prospects are also predicted. Finally, the article is summarized.

## 1. Introduction

With the development requirements of 5G, Internet of Things (IoT), and artificial intelligence (AI) for intelligent and high-performance electronic systems, electronic systems are developing towards high performance, miniaturization, and intelligence, and have been widely used in high-performance computing, smart medical care, autonomous driving, IoT, smart wear and additional devices [1,2,3,4,5]. However, as the miniaturization of feature size gradually approaches the atomic limit, the principle of improving chip performance along Mole’s law gradually fails [6,7,8]. Microsystem technology based on advanced packaging is the latest effective means to promote the development of electronic systems to superior performance and miniaturization [9,10,11]. Vijayara-ghavan et al. [12] proposed a high-performance computing microsystem based on 3D integration technology, integrating a CPU, GPU and DRAM to achieve high throughput and efficient computing. Zaruba et al. [13] used advanced packaging technology to integrate computing cores with high-bandwidth memory into a high-performance memory microsystem with 25% lower power consumption than NVIDA Volta. Burd et al. [14] designed a computing microsystem through advanced packaging technology, with a bandwidth up to 256 GB/s and an energy efficiency of only 1.2 pj/bit. Vivet [15] et al. designed a high-performance computing microsystem based on a variety of processes. The designed microsystem has a 96-core processor and a signal delay of less than 0.6 ns/mm. Fotouhi et al. [16] designed a RF receiving and transmitting microsystem based on three-dimensional integration technology, integrating a coupler, transceiver, array waveguide grating router, etc., which improved the computing performance by 23% and reduced the power consumption by 30%. Based on 3D integration technology, Shulaker et al. [17] proposed a microsystem integrating storage, computing, and sensors to realize high-performance information processing. Tang et al. [18] designed a MEMS gravimeter to achieve extreme sensitivity and a large dynamic range through a suspension design and optical displacement transducer. Yan et al. [19] designed a large capacitance trimethylamine sensor with linear sensitivity in the test concentration range, and developed a prototype sensor based on $Co_{3}O_{4}$@ZnO. Han et al. [20] used a recurrent neural network approach for noise reduction of 3D axial gyroscopes. Gao et al. [21] designed a MEMS filter with a highly robust loan expansion capability by matching the network to broaden and enhance the out-of-band suppression, and applied an aluminum nitride S0 Lamb wave resonator into the filter to improve the loan expansion capability. In the design of microsystems, the high density integration of multiple chips through advanced packaging and high frequency effects, as well as the parasitic differences of different signal paths, lead to difficulties in the design of low-latency group signals. At the same time, in a narrow volume, the parasitic effects of the signal path and the microwave device are more complex and the coupling effects are more pronounced, which makes it difficult to extract the mapping between the signal path design parameters and the signal transmission quality. Park et al. [22] proposed a 192-Gb 896-GB/s 12-high stacked third-generation high-bandwidth memory, and proposed a layout technology based on deep learning to minimize the signal delay deviation, and the proposed method improves the maximum read operation time margin by 33%. Mohammadian et al. [23] designed an optical XOR logic gate based on a ring resonator and a micro-electromechanical system (MOEMS), and established a finite element model of the optical ring resonator to improve the wavelength shift. Rochus et al. [24] proposed a nonlinear mechanical and optical loss of micro-optical mechanical pressure sensor fast modeling method, considering the strong coupling nonlinear mechanics model, and analyzed the location based on the membrane size, residual stress, waveguide, optical wavelength and optical machine coupling effect on the phase rule. TAGHAVI et al. [25] proposed a kind of MOEMS cloth interferometer based on a closed-loop accelerometer, as the the design of closed-loop MOEMS accelerometer has a wider measuring range and higher sensitivity.

The system is greatly reduced in size and integrated with multiple components. Although the system performance is greatly improved, the resulting signal crosstalk, multiscale, multi-field coupling, and other issues make the signal integrity (SI) design more complex and time-consuming. In order to ensure the transmission quality of the key signals of the system, SI has always been the research direction of researchers [26,27]. The interconnect structure and microwave devices are key components for critical signal transmission, and the quality of their SI directly determines the performance of the system [28]. The researchers achieved the goal of high-quality signal transmission by modeling, simulating and optimizing the structure of the interconnection structure and microwave device [29,30,31,32,33]. Approaches to model building fall into two broad categories. The first approach is a detailed model, such as an electromagnetic (EM) model and a finite element model (FEM) [34]. The model is guided by perfect theoretical knowledge, and the established model is extremely accurate, but its computational cost is extremely steep. The second approach is the approximate modeling method [35]. By building empirical models of interconnect structures and microwave devices, or based on equivalent circuit models, the established models simplify the parasitic effects and are quick to compute, but the accuracy is low. As the system frequency increases and the system shrinks, the resulting high-frequency signal crosstalk and multi-field coupling effects exponentially increase the complexity of the SI design. Based on traditional methods such as Monte Carlo, statistics, and worst-case, which additionally exacerbate the shortcomings of EM models and equivalent models [36], in order to improve the efficiency of microsystem SI modeling and simulation, a rapid and accurate microsystem SI analysis method is urgently required.

In recent years, AI, as a modern discipline, has been widely used in performance prediction [37,38,39,40], floor planning [22,41,42], collaborative optimization [43,44,45], image recognition [46,47,48,49], defect detection [50,51,52,53], micromanufacturing processes [54,55] and other aspects of research, and has been successfully applied in microsystem SI design. The application of artificial intelligence methods to microsystem design is commonly divided into four steps [56]: (1) clarify the problem to be solved, determine the design parameters and performance parameters; (2) obtain data; (3) establish a neural networks model and use the acquired data to train neural networks to achieve performance prediction; and (4) optimize the performance through an intelligent optimization algorithm. Among them, performance prediction and performance optimization are two of the most crucial components of AI approaches in microsystem SI design, but a systematic summary of the algorithms and corresponding application scenarios is lacking.

This paper focuses on the performance prediction and optimization design of an AI method in microsystem applications, as shown in Figure 1. The contributions of the present paper are as follows.

(1)The application of NNs in the prediction of SI performance in microsystems are summarized;(2)The application of AI algorithms in the optimization of SI performance in microsystems are summarized;(3)The characteristics and application scenarios of neural network methods applied to microsystem signal integrity performance prediction are compared, and the characteristics and application of artificial intelligence algorithms applied to microsystem signal integrity performance optimization are compared. The above work serves as a reference for an efficient, fast and intelligent microsystem integration design in the future.

The subsequent sections of the paper are arranged as follows. Section 2 mainly introduces the main neural network models applied in the SI design of microsystems, Section 3 mainly introduces the main intelligent optimization methods applied in the SI design of microsystems, Section 4 is the discussion and prospects, and finally, Section 5 is the conclusion.

## 2. Fast Prediction of Microsystem Performance by Neural Networks

Neural networks are working structures similar to the human brain, which can learn the nonlinear mapping relationship between the sampled input and output like the brain [57]. The structure of a typical neural network typically consists of an input layer, hidden layers, and an output layer. In the SI design of microsystems, the input layer is typically the design parameters of the interconnect structure and microwave devices of the microsystem. There are multiple neural cells in the hidden layer, which are mainly used to learn the nonlinear mapping relations between the input and output layers. The output layer is typically a performance parameter that researchers focus on, such as the time-domain response of a microsystem, frequency-domain response, etc. Depending on the complexity of the problem under study, the researcher can adjust the number of hidden layers to adjust the ability of the neural network to learn nonlinear mapping relations. The more hidden layers, the more complex the nonlinear relationship between the input and output. Because neural networks have a very strong nonlinear learning ability, they are widely used in the eye diagram prediction of microsystem interconnection structure [30], crosstalk analysis [37], frequency domain analysis [58], parasitic parameter extraction [59] and other fields. The traditional complex and time-consuming EM or FEM can quickly predict the performance parameters of microsystems [60].

The process of constructing a model for the SI prediction of a microsystem using neural networks is shown in Figure 2. First, the SI design problem of the microsystem needs to be defined, and then the EM simulation model of the microsystem SI needs to be constructed in the HFSS/CST/ADS, and then the design parameters, performance parameters and the range of design parameters are determined according to the needs of the design problem. Then, the type of neural network and the structural parameters of the neural network are determined based on the data characteristics of the type of SI problem, design parameters, and performance parameters. Then, the experimental design method DoE is used to generate the data combination between the design parameters and the performance parameters, the combination of the design parameters and the performance parameters is obtained through the constructed interconnection structure and the EM simulation model of the microwave device, and the data are divided into a training dataset, validation set and test set. Finally, a neural network model trained from the acquired data is used to construct a neural network rapid prediction model for the SI design of microsystems.

Next, neural network architectures that have been successfully applied in the SI design of microsystems in recent years and their examples are discussed.

### 2.1. Artificial Neural Network

The artificial neural network (ANN) is a neural network with the simplest structure. It can learn knowledge of the surrounding environment similar to a brain and store this knowledge in weights [61]. Its predictive function can be written as follows.
(1)$$y=f\left(x\right)=\sum_{j=1}^{M}k_{j}\times G(\sum_{i=1}^{N}w_{ij}x_{i}+b_{j})$$
(2)$$G\left(x\right)=\frac{2}{1+e^{-2x}}-1$$
where *x* is the input vector, $w_{ij}$ is the full-time connecting the $i_{th}$ input node to the $j_{th}$ hidden perceptron, and $k_{j}$ is the weight from the $j_{th}$ hidden perceptron to the output node.

For microsystem design, the prediction of the eye diagram of a high-speed interconnection structure is an essential indicator to evaluate the SI of the microsystem, and numerous researchers have focused on the prediction of eye height and eye width [57,58]. In order to solve the problems of time-consuming model establishment by traditional Monte Carlo method and over-design caused by worst-case design, ANN [57] is applied to the prediction of the eye diagram of an interconnection structure, which is shown in Figure 3a. The network structure is shown in Figure 3b, which consists of three layers, the input layer, the hidden layer, and the output layer. Seven key design parameters are considered in the input layer, namely package impedance $Z_{pkg}$, PCB trace impedance $Z_{pcb}$, transmitter mode upper $Z_{src}$, receiver mode upper $Z_{term}$, driver current $I_{s}$, signal edge rate $t_{r}$, and device capacitance $C_{i}$. The researchers then obtain a set of simulation or test data that can characterize the mapping between the parameters and performance through orthogonal design methods, and train artificial neural networks by obtaining data on the relationship between the parameters and performance. The final input layer has seven nodes, the hidden layer has twelve nodes, and the output layer has two nodes. The eye height and clock jitter errors were trained to be $4.5\times 10^{-5}$. Finally, the trained neural network is used to predict the eye height and clock jitter. The results are shown in Figure 3c. The average test errors for eye height and clock jitter are 0.012 and 0.002, respectively. At the same time, the AI method can quickly predict the eye map and clock jitter without time-consuming circuit simulation.

At the same time, the problem of signal degradation due to crosstalk between high-speed interconnected structures becomes particularly relevant [62,63]. In order to solve the problem that traditional crosstalk analysis requires complete electromagnetic modeling of the signal transmission path and takes a long time to perform time-domain transient simulation, multi-layer perceptron neural networks are applied to the crosstalk prediction of coupled transmission line circuits. First of all, the researchers use ANN to predict the near-end crosstalk of the coupled strip line, four key design parameters are selected, namely conductor spacing S, substrate height H, conductor width W and conductor thickness T, and the output is the near-end crosstalk voltage, then 81 sets of data are sampled by the DoE method to train the established ANN, and the relationship between the four key parameters and the near-end crosstalk is trained by the ANN. The performance of the neural network prediction is R = 0.0075, which indicates that the trained neural network model has a strong generalization ability.

As the frequency increases, the signal crosstalk between different signal transmission lines cannot be neglected. On the other hand, ANN is used to predict the crosstalk at the proximal and distal ends of coupled microstrip transmission lines [37]. First, a physical model of the microstrip transmission line is constructed, as shown in Figure 3d. In the microstrip transmission line, six physical parameters and four properties are chosen as the input and output of the ANN, respectively. The inputs are the substrate height H, conductor thickness T, conductor width W, spacing S between conductors, conductor length L and relative dielectric constant Er of the microstrip transmission line, and the outputs are the maximum near-end crosstalk voltage, the maximum near-end crosstalk occurrence time, the maximum far-end crosstalk voltage and the maximum far-end crosstalk occurrence time. ANN is trained, and the result is shown in Figure 3f–h. The correlation coefficient Rs of maximum near-end crosstalk voltage, maximum near-end crosstalk occurrence time, maximum far-end crosstalk and maximum crosstalk occurrence time are 0.9424, 0.9330, 0.9524 and 0.8896, respectively, indicating that the established neural networks can properly characterize the relationship between the parameters and performance.

The microstrip is the other major transmission structure of the microsystem, and its SI is also crucial for the performance of the microsystem. In order to solve the problem that it takes time to extract the parasitic parameters of the microstrip transmission line model, ANN is applied to the rapid prediction of the RLGC matrix of the microstrip line [59]. As shown in Figure 4a, the electromagnetic model includes three sets of difference pairs. First, it is necessary to determine the modeling parameters, upper and lower bounds, and performance parameters to be extracted. The physical design parameters chosen for the six microstrip designs are the width of the wire *W*, the difference between the pitch *S*, the difference between the different pairs of pitch $S_{p}$, the height of the preg $H_{P}$, the height of the core layer HC, and the relative dielectric constant $D_{K}$, respectively. $L_{o}$, $C_{o}$, $G_{o}$, $R_{o}$, $G_{d}$, and $R_{s}$ are chosen as the key performance parameters in the parasitic parameters of the W model for lossy multi-conductor transmission lines. Since the RLGC matrix of the microstrip is symmetric and reciprocal, the RLGC matrix can be simplified. Only the 11, 12, 13, 23, and 14 components in the RLGC matrix need to be predicted to represent the complete 6×6 RLGC matrix, so the number of output nodes is 30. Then, 150 sets of training data are sampled by LHS to train the ANN. The test graph of the six performance parameters is shown in Figure 4b, and it can be seen that the prediction error of ANN is less than 5%. Ku et al. [64] proposed an ANN method. First, deterministic and random dither components are extracted from the eye images, and then vector fitting techniques are used for preprocessing to reduce the dimensionality of the input data and shorten the training time. The jitter component of the extreme velocity signal can be efficiently separated by training both the eye image and the jitter component.

In the SI design of microsystems, although the general NN method can establish the mapping relationship between the design parameters and performance parameters of microsystems, it does not take into account the inherent physical characteristics and electromagnetic knowledge, which leads to the need for a large amount of data for NNs and reduces the modeling efficiency of NNs. In order to solve the above problems, Chen et al. [65] proposed a knowledge-based NN method to design microwave devices, trained the nine design parameters of the microstrip filter and its S parameters, used prior knowledge as the hidden layer of knowledge neurons, and then trained the NN through the particle swarm optimization algorithm. The microwave filter is designed on this basis. Na et al. [66] proposed an adaptive algorithm for an automatic model structure for knowledge-based parametric modeling. L1-norm optimization is used to automatically determine the mapping in the knowledge-based model. The proposed method is used to design band-stop filters and to reduce the modeling time. Zhang et al. [67] proposed a method combining NN and model order reduction, which solved the problem of pole/zero mismatch in the modeling of microwave passive devices by NN and improved the modeling accuracy. The proposed method was applied to the filter design with an average test error of only 1.37%.

### 2.2. Deep Neural Network

As the SI design problem of microsystems becomes more complex, the learning power of NNs can be improved by increasing the number of hidden layers in order to more accurately capture the nonlinear mapping between the design parameters and performance response. Similar to ANN, the structure of a deep neural network (DNN) is divided into input, hidden and output layers, but the number of hidden layers is increased to h layers. The relationship between the input layer and the h-th layer can be represented by the following formula:(3)$$\left\{z_{j}^{h}\right\}=\left\{x^{h-1}\right\}\left[W^{h}\right]$$
where the $L_{h-1}\times L_{h}$ matrix $\left[W^{h}\right]$ contains the weights from the $h_{1}$ layer to the $h_{th}$ layer, and the output vector $x^{h}$ of the $h_{th}$ hidden layer can be expressed as
(4)$$\left\{x^{h}\right\}=g_{0}(\left\{z^{h}\right\}+\left\{b^{h}\right\})$$
where $g_{0}$ is the activation function. $b_{h}$ is the bias of the $h_{th}$ layer.

In the microsystem SI design, DNN is used to predict the eye diagram of the high-speed channel [29,31,69], and the established high-speed channel model is shown in Figure 4c. The model of the high-speed channel established by DNN is shown in Figure 4d. Eight design parameters are used as input, and the eye height and eye width are used as performance indicators to evaluate the SI of the high-speed channel. The DNN model is trained by collecting data, and the training dataset, validation set and test set contain 717, 48 and 476 data, respectively. The established DNN model has three hidden layers, and the three hidden layers have 100, 300 and 200 nodes, respectively. Finally, the established DNN is used to predict the eye diagram of the high-speed channel of the microsystem. Most of the prediction errors are less than 3%.

Zhang et al. [31] established the high-speed channel model, as shown in Figure 4e. The ten key design parameters are selected as input to the DNN, including the relative dielectric constant *E* of the substrate, conductivity σ and the thickness *H*, the width of the three microstrip lines $w_{1}$, $w_{2}$, $w_{3}$, the thickness $t_{2}$ and $t_{3}$ of the microstrip line conductors 2 and 3, the spacing $s_{1}$ between conductor 1 and conductor 2, the spacing $s_{2}$ between conductor 1 and conductor 3, and the output selected eye height of the microstrip line. Then, the eye diagram of the high-speed channel is analyzed by CST commercial software, and 12372 sets of data are obtained. The ratios of the training dataset, validation set and test set are 70%, 15% and 15%, respectively. The DNN structure adopts two hidden layers, and the corresponding number of hidden units is 12 and 10. The Levenber Marquardt algorithm is selected as the training algorithm. Mean square error (MSE) basically stabilizes and is very small after 20 iterations, indicating that the established DNN model can accurately predict the eye height of the high-speed channel pathway.

Jin et al. [68] predicted electromagnetic interference in wire-bonded ball grid array (WB-BGA) packages using an attention module-based DNN model, and the WB-BGA package model and the proposed model are shown in Figure 4g,f, respectively. The input weights of the DNN are re-derived from a three-layer attention-based module. The input layer is the seven key structural parameters of the package, namely the relative dielectric constant of the center dielectric, the relative dielectric constant of the top and bottom dielectric, the height of the connecting wire, the height of the package cover, the height of the bump, the radius of the signal vias, and the number of ground vias. The output layer is 100 electromagnetic interference radiation values of 0.2–20 GHz. In the final training, 100 nerve cells per layer, for a total of five DNN models, the average relative error is less than 2%, the mean square error MSE is 2.03, and the running time is in the order of milliseconds. The comparison between the DNN model prediction results and experimental measurement results is shown in Figure 4h, and the radiation predicted by the proposed DNN model is in good agreement with the measured results. Jin et al. [70] proposed a novel DNN structure for microwave components, which takes geometric parameters as the input of the multi-layer hiding layer and frequency parameters as the input of the first part of the hiding layer. The proposed structure can reduce the number of training parameters in DNN models and predict the performance of filters through the proposed structure. The maximum number of training parameters is reduced from 1224 to 574, which considerably reduces the training cost.

### 2.3. Recurrent Neural Networks

In the time domain response analysis of microsystem SI, more attention is paid to the analysis of the transmission performance at different time steps. Multiple Newton-style iterations are usually involved at each time step [71], and ANNs and DNNs only focus on the scalar fitting of the output response, thus ignoring the connection between different time nodes in the time-domain response. Recurrent neural networks (RNN) can share weights and feed their outputs back to recurrent inputs, which helps NNs learn the relationship between different time nodes in the time series. An expanded RNN structure with a K step input sequence is shown in Figure 5a, and the structure of RNN is based on a feedback path from output to input. RNN can be expressed as
(5)$$h_{t}=g_{h}\left(x_{t},h_{t-1}\right)$$
(6)$$y_{t}=g_{0}(h_{t})$$
where $h_{t}$ and $x_{t}$ are inputs at hidden state and time t, respectively, and $g_{h}$ and $g_{o}$ are activation functions. However, RNN has the problem of gradient vanishing or gradient explosion. In order to improve the above problems, the long- and short-term memory structure (LSTM) is further proposed. The LSTM network is expressed as
(7)$$i_{t}=\sigma \left(W_{ii}x_{t}+W_{hi}h_{t-1}\right)$$
(8)$$f_{t}=\sigma \left(W_{if}x_{t}+W_{hf}h_{t-1}\right)$$
(9)$$g_{t}=\tan h\left(W_{ig}x_{t}+W_{hg}h_{t-1}\right)$$
(10)$$O_{t}=\sigma \left(W_{io}x_{t}+W_{ho}h_{t-1}\right)$$
(11)$$c_{t}=f_{t}c_{t-1}+i_{t}g_{t}$$
(12)$$h_{t}=o_{t}\tan h\left(c_{t}\right)$$

Nguyen et al. [71,72] proposed a RNN method to generate the time-domain response of the remaining time steps from the SPICE solver, and the PAM2 channel is shown as Figure 5b. Time-consuming transient simulations performed by replacing the SPICE solver can be replaced by simple matrix-vector multiplicative inference, further reducing the computation time. By sampling three voltage transient signals and dividing the normalized sampled signals into sequence blocks of length *k*, the transient behavior of the circuit was solved by using a SPICE emulator. Through RNN training the sampled sequences, the established NNs model consists of four LSTM units, and each LSTM unit has 20 hidden units. Through the established RNN model, how to accurately and efficiently predict the $R_{X}$ voltage, and the prediction voltage at the receiver with LSTM are shown in Figure 5c, which shows that the voltage can be predicted accurately.
(13)$$y=f\left(h*x\right)=f\left(\sum_{m=-\infty }^{m=\infty }x\left[n\right]h\left[m-n\right]\right)$$

### 2.4. Convolutional Neural Network

The frequency domain response is another important aspect of the microsystem SI design, and the use of ANN, DNN and other fully connected networks for training discrete frequencies. CNN, by the convolution layer, pooling layer, fully connected layer, has the main role of feature extraction, downsampling, and classification to minimize the loss of NN by the layer down function weight value layer by layer inverse adjustment, to further improve the accuracy of network training.

Ren [73] et al. proposed a NN model based on a convolutional autoencoder (CAE), as shown in Figure 5d. The mapping relationship between the image feature and S parameter of the planar filter is learned by the encoder and dense layer. CAE modeling steps are as follows: (1) determine the shape of the filter, frequency range of S parameter and target; (2) generate data sets, and obtain different coupling matrices to generate S parameters by changing the design parameters such as length and gap; (3) train the model, split the low-pass filter image into $P_{x}\times P_{y}$ pixels, and extract the encoder through the unsupervised learning training model; and (4) connect the dense layer to the encoder, and construct the dense layer through transfer learning. The comparison results between the traditional CNN and the proposed convolutional encoder are shown in Figure 5e. The average calculation error of the traditional CNN model is $4.5\times 10^{-2}$, while that of the proposed convolutional encoder is $1\times 10^{-2}$, indicating that the proposed convolutional encoder can effectively improve the accuracy of the prediction model.

Torun [74] et al. proposed a spectrum transposed convolution network (S-TCNN) to solve the problems of the large number of geometric parameters, low design efficiency and low training efficiency in the design of solenoid inductors. The proposed S-TCNN architecture is shown in Figure 6a and the geometry of the solenoidal inductor is shown in Figure 6b. It uses a one-dimensional Kernel to achieve a feature extraction of the frequency axis with a normalized mean square error (NMSE) of 15.2%. A comparison of S-TCNN and FC-NN to EM simulations is shown in Figure 6c, and realizes accurate modeling of the core solenoid inductor with a small amount of data.

Then, Torun [75] et al. used the S-TCNN to predict the frequency response of large bandwidths, and added a causal execution layer (CEL) and a passive execution layer (PEL) to improve the causality and passivity of the model. The Block diagram summary of the operations performed in CEL and PEL is shown in Figure 6d. The model of the differential PTH pair and differential BGA pair is established, which is shown in Figure 6e. The input layer is the structural parameter, and the output layer is the S parameter, which can realize the prediction of the S parameter of 0.1–100 GHz and the step size is 100 MHZ, and the normalized mean square error is less than 6%. The Block diagram summary of the operations performed in CEL and PEL is shown in Figure 6f, and the proposed method can ensure the passivity of prediction results.

### 2.5. Summary

NN-based prediction methods have been widely used in SI prediction. The comparison of SI prediction algorithms is shown in Table 1. At present, ANNs and DNNs are still the dominant ones, and they are widely used in the analysis of microsystem interconnection structures, microwave devices in the time-domain, and cross-talk. Due to the limited number of hidden layers in ANN, it is suitable for microsystem SI prediction with a simple relationship between design parameters and performance parameters. DNN has a higher number of hidden layers than ANN, so it can predict the signal completeness with a complex relationship between the design and performance parameters, but the amount of data used for training also increases. These two NNs only focus on error reduction on scalar data and do not reflect the physics of SI prediction in microsystems. For high-dimensional prediction in the time and frequency domain, reducing the dimensionality of the high-dimensional input data through data preprocessing and then training with ANN and DNN is an effective approach. On the other hand, NN structures that can capture the relevance of input data can be used for prediction. RNN and CNN can capture the time-domain and frequency-domain correlations of the input parameters, respectively. Therefore, RNN and CNN are suitable for performance prediction of high-dimensional problems with time and frequency domain responses, respectively, and can increase the extrapolation capability to some extent. In addition, STCNN + CEL + PEL can further ensure the causality and passivity in performance prediction and reduce the amount of data required for training due to the addition of the causal forcing layer and passive layer. Although it is possible to improve the performance of NN compared to traditional methods, from design parameters to the speed of mapping between them, the direct use of classical NN methods still incurs a large training cost. By conducting an in-depth analysis of the SI problem to be solved, the NN performance prediction method can be built with relevant knowledge to reduce the training cost and improve the extrapolation ability of the NN. Fast performance prediction methods for microsystem SI based on NNs offer the possibility of intelligent optimization of SI.

## 3. Intelligent Optimization Method for Microsystem Design

The previous sections mainly describe the rapid prediction of interconnection structure and microwave device design performance by AI methods during the SI design of microsystems without full-wave or circuit simulation. However, if the SI performance of the microsystem does not meet the requirements, the performance parameters of the microsystem must be optimized [31]. With the shrinking of the size of the microsystem and the increase in the frequency of signal transmission, the influence of the design parameters of the interconnection structure and the microwave device on multiple performance parameters is more complicated [76]. In addition, the trade-off between different performance parameters is also more complicated; that is, the improvement of one performance parameter will lead to the degradation of additional performance parameters [32]. Therefore, traditional optimization methods based on empirical knowledge and statistical methods need to undergo a time-consuming trial and error process [31], and the robustness of the optimization crosses. However, the heuristic optimization algorithm based on AI has been widely used in the interconnection structure of microsystems and the optimization design of microwave devices in recent years because of its strong global and local search capabilities [29,31,59], which considerably improve the design and optimization efficiency of SI in microsystems.

The process of using an AI method to optimize the microsystem is shown in Figure 7. First, it is necessary to determine the optimized design parameters and performance parameters and initialize the algorithm, and then design the optimized target function according to the performance parameters that need to be optimized, and it is used as the fitness function of the optimization algorithm. The fitness function of the algorithm is then used to evaluate the quality of the individual performance parameters via the established fitness function. Finally, the optimization ends when the error between the fitness function corresponding to the optimized performance parameter and the globally optimal fitness function is smaller than a set accuracy threshold or the number of iterations reaches a preset value.

Next, the SI optimization algorithms applied in microsystems in recent years will be summarized.

### 3.1. Genetic Algorithm

The genetic algorithm (GA) is designed and proposed in accordance with the laws of biological evolution in nature, simulating the natural selection of Darwin’s theory of biological evolution and computational models of the biological evolution processes of genetic mechanisms. Original individuals are intelligently selected through crossover and mutation operators until the average fitness function and the maximum fitness function converge to achieve efficient optimization.

Step 1: Initialization. Initialize evolutionary algebra and groups;

Step 2: Evaluate individuals. Evaluate the quality of individuals in the group according to the moderate function;

Step 3: Selection. Select the dominant individual in the population;

Step 4: Crossover. The chromosome parts of the selected dominant individuals are swapped to create new offspring;

Step 5: Mutation. Make random changes in certain chromosome values of the generated new individual;

Step 6: Algorithm termination condition. The number of iterations reaches the maximum number, or the difference between the current fitness value and the optimal fitness value is less than the set threshold.

At present, GA is applied in the crosstalk optimization [59] and eye height [31] of the microsystem interconnection structure. In [59], the GA optimization algorithm is applied to optimize the loss and crosstalk of the transmission line. First, the relationship between the design parameters and performance parameters of the transmission line is quickly predicted through the established ANN model, and then 6 are selected. The design parameters and 5 performance parameters are optimized, namely differential mode impedance ($Z_{DIFF}$), common mode impedance ($Z_{COMM}$), attenuation constant (*a*), near-end crosstalk (NEXT) and far-end crosstalk (FEXT). The optimization is carried out by using the linear accumulation of errors between multiple performance parameters and the target design specification as the fitness function of the optimization function. The optimized *W*, *S*, $S_{P}$, $H_{P}$, $H_{C}$, and $d_{k}$ are 5.2 mil, 10 mil, 50 mil, 4.9 mil, 4.3 mil, 3 mil, respectively. Finally, the optimized design parameters are brought into the electromagnetic simulation model of the CST to verify the performance parameters, and the verification result is shown in Figure 8a. Zhang et al. [31] built a fast prediction model of an eye map by DNN, and the structural model is shown in Figure 8b. Then, ten key design parameters are jointly optimized using the GA, and the best fitness value is 0.999. Then, CST software is used to verify the eye map, and the result is shown in Figure 8c. In addition, the eye height is 0.998. Therefore, GA can efficiently perform the collaborative optimization of multiple parameters. Zhu et al. [77] proposed a kind of ANN and GA wire-bonding interconnect performance optimization method, with the optimal time reduced to 0.2 from 7.63 h, and improved the optimization efficiency. Odaira et al. [78] created an eye diagram based on the GA optimization method, where the proposed methods were be elevated and eyes width increased 3.07 times and 1.06 times, respectively.

### 3.2. Differential Evolution

The differential evolution (DE) algorithm [82] is a global evolutionary algorithm that aims to find the design variable *x* with the maximum estimated return rate. Its optimization process is divided into four steps [83].

Step 1: Initialization of the parameter vectors.

Step 2: Mutation. Subtract the two candidate design parameters to generate a difference vector, which is then weighted and added to the third candidate design. The mutation vector can be expressed as
(14)$$V_{i}\left(g+1\right)=X_{r1}\left(g\right)+F\left(X_{r2}\left(g\right)-X_{r3}\left(g\right)\right)$$
where $r_{1}$, $r_{2}$ and $r_{3}$ are random integers of mutual exclusion. $F_{s}$ is the scaling factor, which represents the differential vector weights.

Step 3: Cross. The mutation vector $v_{i}$, *G* and the target vector $x_{i}$, *G* are recombined to form a new test vector $u_{i}$, *G*

Step 4: Selection. Choose the better of the estimated values between the test vector and the target vector.

In order to solve the problems of poor optimization quality and low efficiency due to the increase of sensitive design parameters when using traditional optimization methods to optimize filter performance, Zhang et al. [79] used the DE algorithm to optimize filter performance benefits. First, for the x-band filter with 11 sensitive design variables (as shown in Figure 8d), the sensitive design parameters are x = [$l_{1}$, $l_{2}$, $l_{3}$, $l_{4}$, $l_{5}$, $k_{12}$, $k_{23}$, $k_{34}$, $k_{45}$, $q_{e1}$, $q_{e2}$], the constraint is 2 between the operating frequency of 9.8 GHz–9.85 GHz, and the pass rate is estimated to be 47.5% through 40 electromagnetic simulations. Therefore, a fast prediction model of the filter is first established through a radial basis function NN, the S-parameters are optimized by DE Algorithm, the optimized design parameters with the highest pass rate are: x* = [18.941, 19.991, 19.817, 19.749, 18.770, 4.383, 4.387, 5.516, 4.977, 8.465, 8.958] mm, and the pass rate can reach 82.5%. Compared with the traditional optimization method, the pass rate of the filter design can be greatly improved. Then, use the DE Algorithm to optimize the design parameters of the C-band bandpass filter. The model of the established bandpass filter is shown in Figure 8e. Fourteen sensitive design parameters are selected for optimization. The optimized pass rate can reach 97.2%, which is 41.2% higher than the traditional method. Therefore, the DE algorithm can effectively optimize the performance of the filter.

### 3.3. Deep Partition Tree Bayesian Optimization

The deep partition tree Bayesian optimization (DPT-BO) algorithm is a high-dimensional global optimization method proposed to address the problem that the number of simulations required for convergence of traditional BO algorithms grows exponentially with the number of optimization parameters [80], and is therefore not suitable for high-dimensional optimization. When training an additive Gaussian process model, DPT-BO uses full additivity decomposition to consider the interaction between the parameters, which makes the algorithm more suitable for high-frequency design optimization. By using a different deep partition tree approach, the auxiliary optimization step in the BO algorithm is eliminated and the high-dimensional sample space can be covered quickly with fast convergence. The procedure of the deep partition tree Bayesian optimization algorithm is as follows.

Step 1: Enter the sample space.

Step 2: Train an additive Gaussian process model.

Step 3: Group input parameters according to their sensitivity to $f\left(x\right)$.

Step 4: PI, EI and UCB are used as acquisition functions to avoid bias.

Step 5: Using the deep hierarchical partition tree method, the region is expanded in the vertical direction and then partitioned in the horizontal direction to generate additional candidate points.

Step 6: Select the candidate that maximizes the value of the obtained function as the sampling point and evaluate the objective function value at the sampling point. If the target requirement is met, the optimal value is output. Otherwise, the loop continues.

The DPT-BO optimization algorithm [80] was applied to optimize the SI of the microstrip line, and the structure of the microstrip is shown in Figure 8f. First, the surrogate model of the microstrip is built using the GP method, and then ten control parameters are optimized by the DPT-BO. The GP model is used to determine the RLGC matrix for each cell of the microstrip, which is converted into an S-parameter and then cascaded to form an interconnect channel of the length 10 mm. ADS generates an eye map by bit-by-bit simulation using the channel’s S-parameter, and the generated eye width and eye height are fed back to the optimization algorithm for the next iteration. A performance comparison between the DPTBO algorithm and other algorithms is shown in Figure 8g, and the proposed algorithm converges quickly in a high-dimensional design. After optimization with DPT-BO, the eye width and eye height are 53.13 ps and 0.54 V, respectively. Additionally, the internal jitter is 8.12 ps, and the convergence rate is 1.41 times, 1.48 times and 1.19 times faster than that of TSBO, respectively.

### 3.4. Two-Stage Bayesian optimization

The two-stage Bayesian Optimization (TSBO) algorithm uses two phases of optimization [84], namely the rapid exploration phase and pure development phase, to reduce the number of simulations needed to find the optimal design point, and thus reduce the computational overhead. The process of the two-stage Bayesian optimization algorithm is divided into two stages. In the first stage, the region containing the global optimal is quickly found in the sample space, and the optimal collection function is determined. In the second stage, the optimal acquisition function is used in the region selected in the first stage to fine-tune the optimization problem and improve the accuracy, and the specific algorithm steps are as follows:

Step 1: Enter the sample space.

Step 2: Divide the sample space X into a two-dimensional hyperrectangular region to generate candidate points.

Step 3: Use PI, EI, and UCB as fetch functions in sequence. After obtaining a specified number of observations, the algorithm exits the sequential strategy and selects the method with the maximum gain as the acquisition function.

Step 4: Select the candidate point that maximizes the value of the obtained function as the sampling point; evaluate the objective function value of the sampling point; and select the region where the sampling point resides as the new region.

Step 5: Output the current optimal value, and enter the second stage optimization when the Euclidean distance between the current sampling point and the previous sampling point is sufficient (phase switching standard).

Step 6: Carry out a more detailed regional division within a small enough region optimized in the first stage to generate candidate points.

Step 7: Select the candidate point that maximizes the value of the obtained function as the sampling point; evaluate the objective function value of the sampling point; and select the new area.

Step 8: Output the current optimal value and continue the loop until the target requirement is met.

TSBO is applied to the collaborative optimization of the clock deviation and temperature gradient to improve the SI of 3D integrated circuits. A total of five control parameters are considered [81]. The temperature gradient optimized by TSBO is 23.5 ${}^{\circ }$C and the clock deviation is 86.0 ps, which are both better than IMGPO and the nonlinear solver. The convergence rate of TSBO for the lowest temperature gradient is 3.76 times faster than IMGPO and 3.96 times faster than the nonlinear solver, respectively. TSBO is also used for the multi-objective collaborative optimization of the integrated voltage regulator (IVR) [81], and the two-chip SiP IVR architecture is shown in Figure 8h. Ten control parameters are used to optimize the two objectives of maximizing the power efficiency of the integrated voltage regulator and minimizing the embedded inductance region. The inductance size is determined by TSBO, and then input into the full-wave solver (Ansys HFSS Ver. 2015.2., Ansys Maxwell Ver. 2015.2.) to extract the two-port Z matrix. Inductance and a previously developed step-down converter model are then used to calculate the IVR efficiency. The calculated efficiency is combined with the inductance region and fed back to the TSBO for the next iteration. After TSBO optimization, the peak efficiency of IVR can reach 85.1%. The embedded electromagnetic inductor covers an area of 5.1 mm${}^{2}$, resulting in a 5.7% increase in efficiency and a 56.1% reduction in area compared to the manually tuned design. Moreover, TSBO reduces the CPU time required to complete the optimization by 72.4%, 57.4% and 56.7% compared to the nonlinear solvers GP-UCB and IMGPO, respectively.

### 3.5. Summary

Optimization methods based on evolutionary algorithms have been widely used to optimize SI in microsystems. Most of the aforementioned optimization methods are based on established NN prediction models, which can accelerate the iteration speed. A comparison of specific optimization algorithms is shown in Table 2. Evolutionary algorithms such as GA and DE have been proposed earlier and are more mature and, therefore, less difficult to apply. Currently, they have been applied to eye image, crosstalk, and filter optimization, and the efficiency and effectiveness of the optimization have been improved to some extent. However, due to the structural nature of genetic algorithms, the number of populations and the running time are exponentially large and, hence, computationally slow in the case of increasing populations or high-dimensional optimization. DPTBO uses the deep partition number method to quickly cover the high-dimensional space, which improves the ability of Bayesian optimization algorithms in high-dimensional problems. Thus, DPT-BO is more suitable for the collaborative optimization of high-dimensional design parameters, but its structure is complicated. TSBO accelerates its convergence by splitting the optimization part into fast exploration and optimization phases, but the high-dimensional problem limits its application.

## 4. Discussions and Outlook

It can be seen from the design example of SI discussed in this paper that AI approaches have been widely used in the field of performance prediction and optimization of SI in microsystems. In the context of SI prediction in microsystems, NNs are the main AI methods, which are mainly used in high-speed signal path-eye map prediction, crosstalk prediction, parasitic parameter prediction, frequency response prediction, etc. Using the obtained data to train a NN, the traditional electromagnetic/circuit simulation model is replaced by NN, which greatly improves the efficiency of the simulation. For different application scenarios, the NN structure suitable for the problem should be selected based on the characteristics of different NNs. The architecture of ANN is simple and therefore suitable for SI prediction in microsystems with relatively simple input and output quantities. DNN increases the number of hidden layers on top of ANN, thus increasing the ability to map between the design and performance parameters and improving the prediction accuracy. However, additional training data are needed to determine the weights between different layers, which increases the training cost. ANN and DNN only focus on the scalar patchwork between the design parameters and performance and do not reflect the correlation between performance in the time and frequency domain. The RNN constructs the correlations before and after the time domain by adding feedback paths, and the CNN constructs the correlations between different frequency points by convolutional layers. Therefore, these two types of NNs are suitable for performance prediction in both time and frequency domains. In addition, when predicting performance, prior knowledge can be added to the hidden layer of NN to reduce the amount of training, and ensure the electromagnetic characteristics of the structure and device itself. Following the development of fast predictive models of NN performance, heuristic optimization algorithms have been widely used in the optimization of SI in microsystems. Classical algorithms such as GA and DE are relatively mature, have strong optimization robustness, and are robust in the optimization process for low dimensional parameters. The DPTBO method combines deep partition trees to quickly traverse the high-dimensional design space, and thus has a clear advantage in high-dimensional optimization problems. TSBO splits the optimization problem into two stages, which can quickly locate the target region and then accurately search for the optimal solution. Hence, it has a significant impact on the need for fast convergence. The combination of fast prediction models and optimization algorithms for microsystem SI can replace traditional simulation models based on electromagnetism and optimization methods that rely on expert experience or statistics, and considerably increase the efficiency of design and optimization.

In the future, the system volume will be further reduced, multi-field coupling effects will be more severe, and the trade-off relation between multiple software iterations and multiple performance metrics will be complicated, which will lead to a lower efficiency when using traditional analysis methods. In addition, the meshing and solution times will be further improved when the multi-scale components are integrated in microsystems. AI methods may be an effective approach to solve the above problems. By solving for the weights of the hidden layers, the flow of data from the design parameters to performance metrics during the multi-software iteration can be constructed to reduce the design difficulty. In addition, NN models can be constructed to skip steps such as grid partitioning and time-consuming steps due to cross-scale effects, thus improving the simulation efficiency. Although the AI approach can considerably improve the design efficiency of microsystem SI, various challenges remain. (1) **Extreme training cost.** The accuracy of NN training is closely related to the number of samples, and electromagnetic/circuit simulations are still required to obtain the data, which still consume a significant amount of the training cost. Although the DoE method can reduce the number of combinations of the acquired data to a certain extent, it still cannot determine the minimum amount of training data required to ensure the accuracy of the predictive model. (2) **Extrapolation ability.** Currently, most NNs have excellent predictive performance within the training set, but poor predictive performance, that is, poor extrapolation ability, once the design parameters jump out of the design space. Improving the extrapolation performance can reduce the training cost to some extent and further reduce the design cycle. (3) **Reverse design.** Currently, most studies focus on fast prediction of the corresponding performance parameters through design parameters. However, in practical engineering problems, the performance requirements are generally known first and the design parameters need to be addressed. Therefore, it is more important to investigate the inverse design methods to solve engineering problems. (4) **High-dimensional optimization problem.** In complex microsystems, the coupling effects between the parameters are more pronounced, and the relationships between multiple design parameters are more complex. Therefore, dimensionality reduction optimization methods should be studied or AI methods suitable for higher dimensions should be developed to shorten the design cycle of microsystems.

## 5. Conclusions

This paper introduces the application of AI technology in microsystem SI performance prediction and optimization design, and summarizes and compares the characteristics of the main NNs methods of performance prediction and their application scenarios in microsystem SI design. Then summarizes and compares the characteristics of optimization design methods and application scenarios in microsystem SI optimization design. Finally, different prediction algorithms and optimization algorithms are discussed and compared. The main conclusions are as follows:1.NNs can be used to quickly predict the SI of microsystems, but to ensure the accuracy of the prediction, a large amount of data needs to be obtained to train NNs.2.The SI prediction problem with independent design parameters, a small number of design parameters and performance parameters, and a relatively simple mapping relationship can generally be solved by NNs such as ANN or DNN; if there is a certain correlation between the design parameters, RNN or CNN can be selected. Problems that have a certain physical significance and need to ensure that the constructed network has physical properties such as causality and passivity must add relevant knowledge according to the specific problem as a priori to ensure its characteristics.3.The heuristic optimization algorithm can improve the optimization efficiency of the optimal SI solution, and the combination of the established fast prediction model based on NN can further reduce the iteration time.

## Figures and Tables

**Figure 1 micromachines-14-00344-f001:**
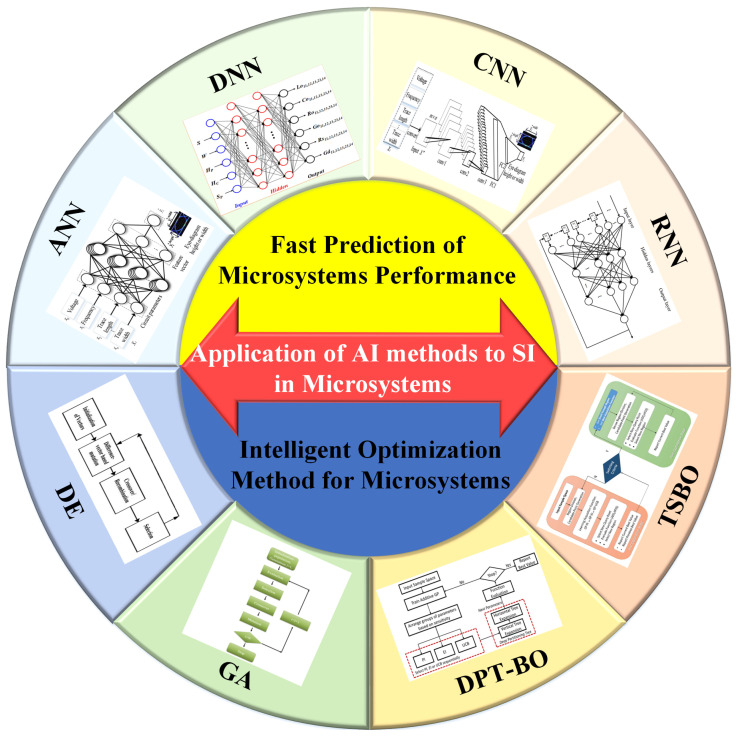
AI methods of SI design in microsystems.

**Figure 2 micromachines-14-00344-f002:**
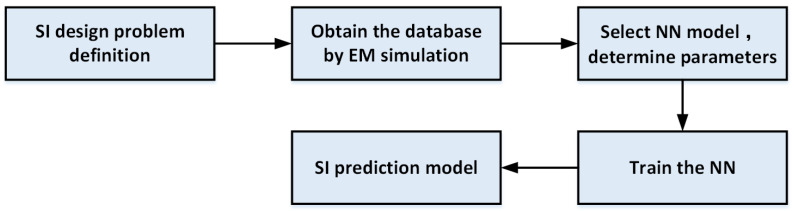
Flow chart of SI prediction model of microsystem constructed by NNs.

**Figure 3 micromachines-14-00344-f003:**
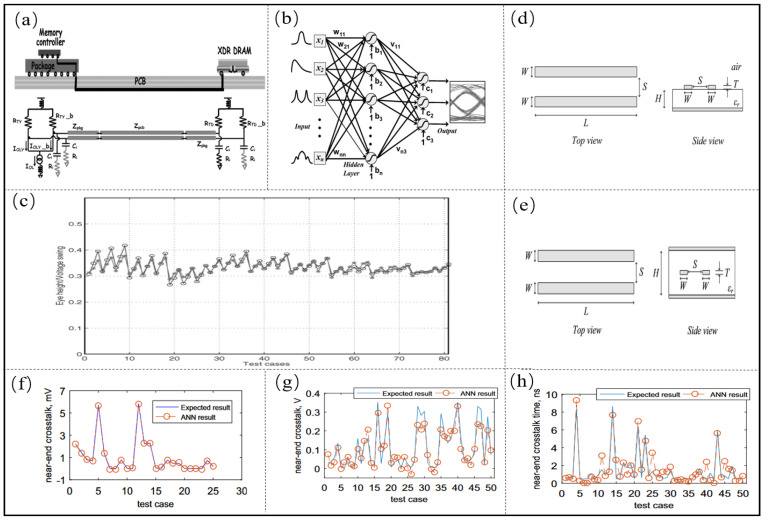
(**a**) Short-channel XDR memory system. Adapted with permission from [57]. (**b**) Established ANN model. Adapted with permission from [57]. (**c**) Eighty-one test cases to verify the approximation capabilities of the ANN. Adapted with permission from [57]. (**d**) Geometrical structure of a coupled stripline. Adapted with permission from [37]. (**e**) Geometrical structure of a coupled stripline. Adapted with permission from [37] (**f**) Testing result of the ANN for crosstalk in striplines. Adapted with permission from [37]. (**g**) Maximum near-end crosstalk voltage comparison results. Adapted with permission from [37]. (**h**) Comparison of maximum near-end crosstalk occurrence time results. Adapted with permission from [37].

**Figure 4 micromachines-14-00344-f004:**
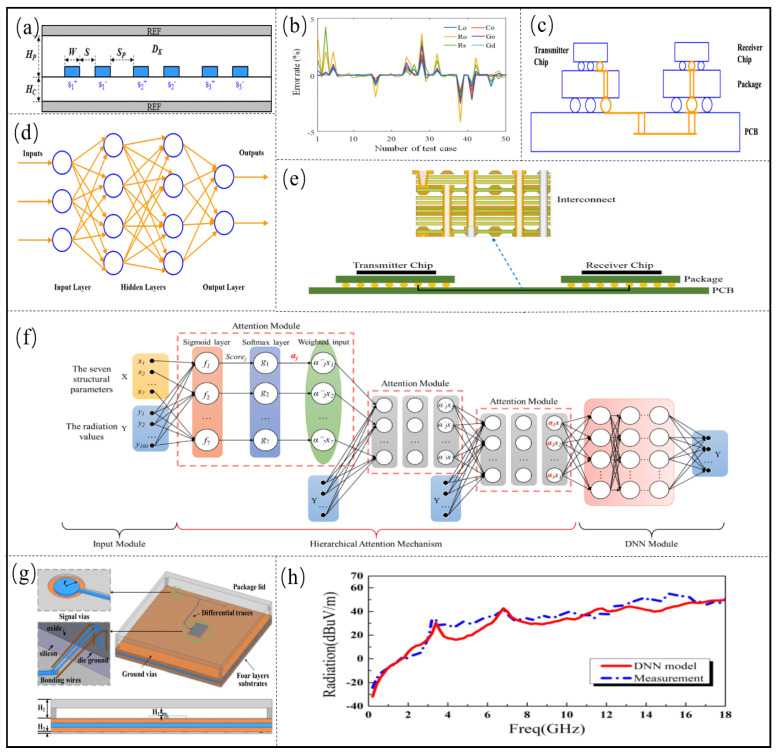
(**a**) Target channel for modeling. Adapted with permission from [59]. (**b**) Validation of predicted channel RLGC by ANN model. Adapted with permission from [59]. (**c**) The topology of a high-speed channel. Adapted with permission from [29]. (**d**) Model of high-speed channel established by DNN. Adapted with permission from [29]. (**e**) Cross-section view of the high-speed channel. Adapted with permission from [31]. (**f**) Hierarchical attention-based DNN. Adapted with permission from [68]. (**g**) WB-BGA package model. Adapted with permission from [68]. (**h**) Comparison of EMI radiation results predicted by DNN model and results measured by far-field experiment. Adapted with permission from [68].

**Figure 5 micromachines-14-00344-f005:**
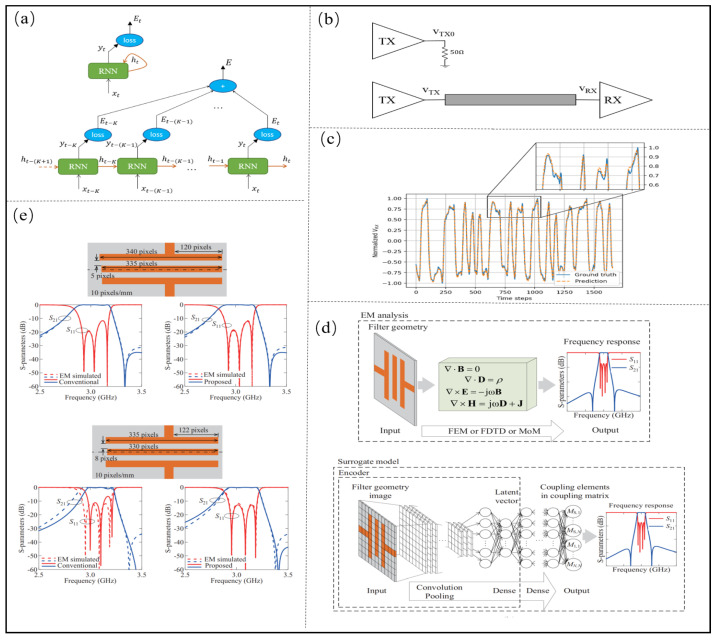
(**a**) Unrolled RNN with an input sequence of K steps. Adapted with permission from [71]. (**b**) PAM2 channel. Adapted with permission from [71]. (**c**) Predicted voltage at the receiver VRX with a LSTM network. Adapted with permission from [71]. (**d**) Proposed CAE-based CNN model. Adapted with permission from [73]. (**e**) The comparison results between the traditional CNN and the proposed convolutional encoder. Adapted with permission from [73].

**Figure 6 micromachines-14-00344-f006:**
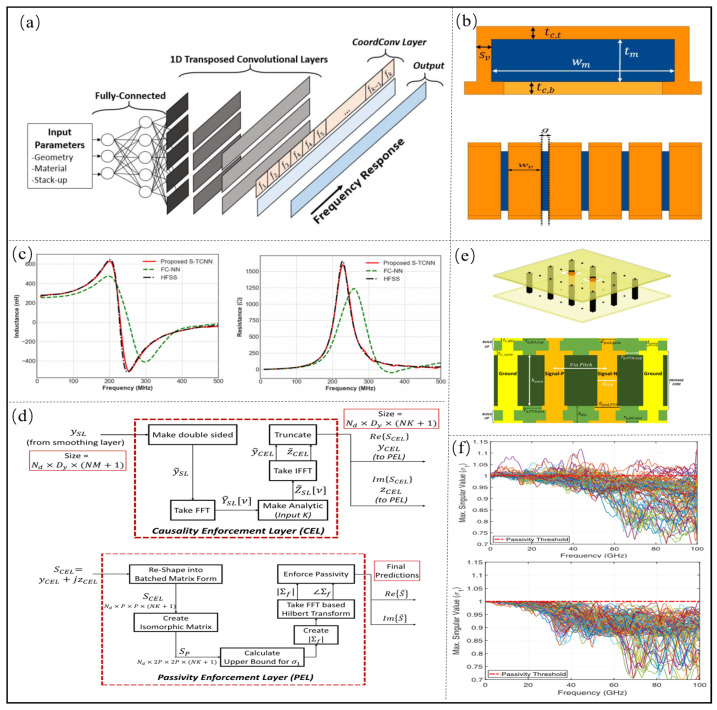
(**a**) Proposed S-TCNN architecture. Adapted with permission from [74]. (**b**) Geometry of the solenoidal inductor. Adapted with permission from [74]. (**c**) Comparison of S-TCNN and FC-NN to EM simulations. Adapted with permission from [74]. (**d**) Block diagram summary of the operations performed in CEL and PEL. Adapted with permission from [75]. (**e**) Geometry of the differential PTH structure. Adapted with permission from [75]. (**f**) Passivity characterization of the predicted S-parameters. Adapted with permission from [75].

**Figure 7 micromachines-14-00344-f007:**
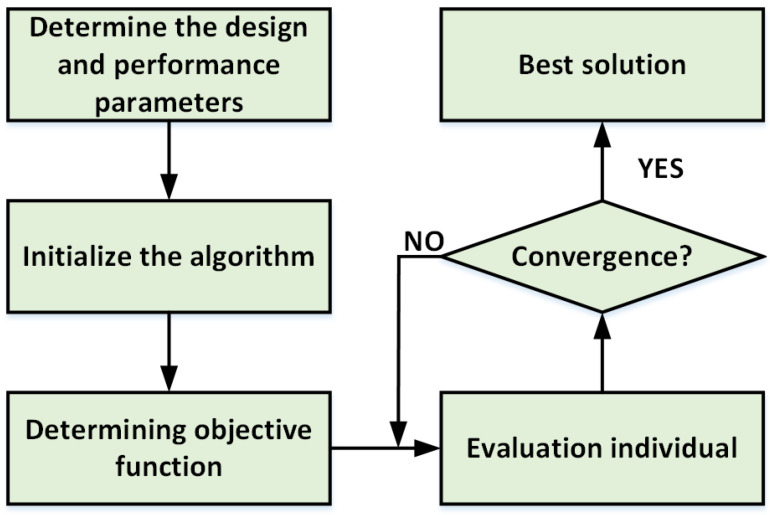
The process of using an AI method to optimize the microsystem.

**Figure 8 micromachines-14-00344-f008:**
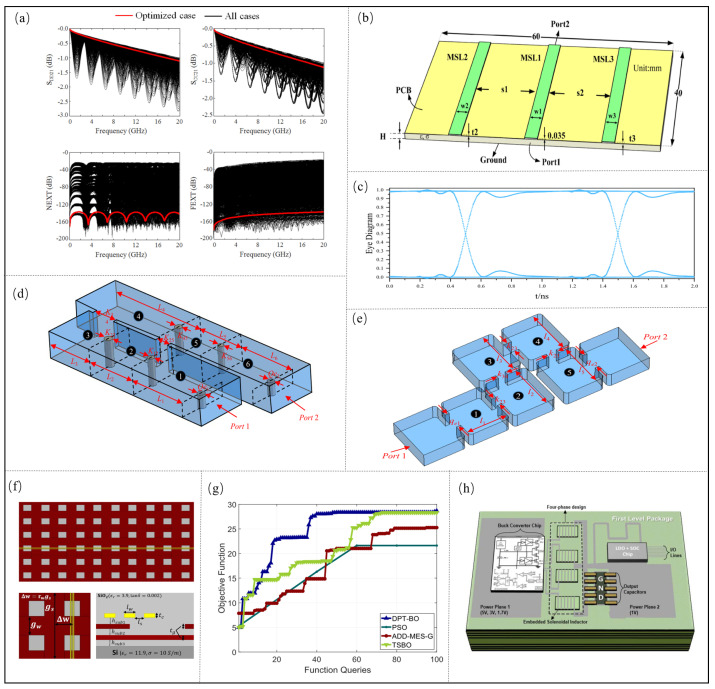
(**a**) Validation of channel optimization result by GA. Adapted with permission from [59]. (**b**) Structure and dimension of a high-speed channel. Adapted with permission from [31]. (**c**) Eye diagram obtained by the CST commercial software based on the optimized ten parameters. Adapted with permission from [31]. (**d**) Structure of x-band filter. Adapted with permission from [79]. (**e**) Structure of bandpass filter. Adapted with permission from [79]. (**f**) Structure of the high-speed channel. Adapted with permission from [80]. (**g**) Performance comparison between DPTBO algorithm and other algorithms. Adapted with permission from [80]. (**h**) Two-chip SiP IVR Architecture. Adapted with permission from [81].

**Table 1 micromachines-14-00344-t001:** Comparison of SI prediction algorithms.

Ref.	Application Fields	Design Variables	Methods	Passivity, Causality	Advantage	Deficiency
[30]	Predicted channel eye height and jitter	5	ANN	No	High speed	Requiring a large amount of data and fewer design variables
[37]	Predicted the crosstalk of coupled strip line and microstrip	4–6	ANN	No	High speed	Requiring a large amount of data and fewer design variables
[59]	Predicted channel loss and crosstalk	6	ANN	No	High speed	Requiring a large amount of data and fewer design variables
[29]	Predicted channel eye height and eye weight	8	DNN	No	High accuracy	Requiring a large amount of data
[31]	Predicted channel eye height	10	DNN	No	High accuracy	Requiring a large amount of data
[68]	Predicted the maximum 3m radiated electric field	7	Hierarchical attention-based DNN	No	High accuracy and low cost	Fewer design variables
[71]	Predicted the voltage waves	3	RNN	No	Strong extrapolation ability	Gradient disappears and gradient explodes
[73]	Predicted the S-parameter of BPF	4	CNN	No	Processing high dimensional data	Requiring a large amount of data
[74]	Predicted the inductance	8	STCNN	No	High speed, accuracy, and require less data	Poor physical consistency
[75]	Predicted the frequency response of PTH pair and BGA pair	8–13	STCNN + CEL + PEL	Yes	High accuracy, physical consistency, and requiring a small amount of data	Lower speed
[65]	Predicted the frequency response of microstrip hairpin filter	6	ANN + Knowledge	Yes	High accuracy, and requiring a small amount of data	Requiring the knowledge
[66]	Predicted the frequency response of microstrip filter	7	ANN + Knowledge	Yes	High accuracy, and requiring a small amount of data	Requiring the knowledge
[67]	Predicted the frequency response of three-pole H-plane filter	9	ANN + Knowledge	Yes	High accuracy, and requiring a small amount of data	Requiring the knowledge

**Table 2 micromachines-14-00344-t002:** Comparison of SI optimization algorithms.

Ref.	Application Fields	Number of Optimization Parameters	Methods	Advantage	Deficiency
[59]	Optimize channel loss and crosstalk	6	GA	High robustness and simple structure	Small optimization dimension and slow convergence
[31]	Optimize the eye height	10	GA	High robustness and simple structure	Small optimization dimension and slow convergence
[79]	Optimize the pass rate of filters	11, 14	DE	High robustness and simple structure	Small optimization dimension
[80]	Optimize the eye diagram, S parameters and WPT	10, 14, 32	DPTBO	High optimization dimension	Complex structure
[81]	Optimize the clock deviation and temperature gradient	10	TSBO	Fast convergence	Small optimization dimension

## Data Availability

Not applicable.

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
