# Peer review of "Application and Prospect of Artificial Intelligence Methods in Signal Integrity Prediction and Optimization of Microsystems"

_micromachines, 2023, doi:10.3390/mi14020344_

Round 1

Reviewer 1 Report

The signal integrity topic of this review manuscript is of interest, in particular the discussion on the hot topic of semiconductor device reliability – the data eye diagram. The followings are some major points to improve the quality of the current manuscript.

1. There is a typo on Page 2, Line 63 in the manuscript, “conclution”.

2. The authors addressed an important yet interesting problem. From the manufacturing viewpoint, the signal integrity is the de facto critical bottleneck of DDR5 data lines/buses beyond 4800 mega-transfers per second. From the memory signal integrity perspective, it would be better to add an in-depth detailed discussion on the cutting-edge on-DIMM DDR5/DDR4 bus signal integrity and the eye diagram (on the prediction of eye height and eye width) of low-latency and high-speed DRAM microsystems, both at the DRAM chip level and the DIMM circuit board level. Example vendors include Changxin Memory Technologies, Micron, SK Hynix, and Samsung. The readers will be more interested with the manuscript. Theoretical analysis is expected to appear in the manuscript for publishing in Micromachines.

3. The eye diagram is too small. It would be better to draw the eye diagram in Figure 3 larger to make the manuscript more intuitive.

4. Why the authors did not consider the parasitic capacitance of transmission lines in their latency and power model?

5. The manuscript is full of abbreviations and the readability is poor. Please reduce that and use complete spelling, both in the figures and in the main text.

6. It would be better to summarize the training algorithm of the high-dimensional advanced neural networks. It seems that the authors only reviewed general methods. The domain-specific optimization of neural networks is not clear in the manuscript.

7. Multiple typos occur in Table 1: “Predicte”.

8. It would be better to organize the manuscript in terms of the optimization goals/performance metrics, i.e., the application fields in Table 1 the authors claimed.

9. I guess that the “GA” in subsection 3.1 means the genetic algorithm. The authors should use complete spelling in their manuscript.

10. The insights derived from this review manuscript is relatively weak. The reviews on electrical-mechanical microsystems and sensor microsystems are missing.

11. The references section only has 50+ citations and is not adequate enough. Otherwise, the significance of this review manuscript is not up to the mark. Typical review papers are expected 200+ references for an essential readership.

Author Response

Ms. Ref. No.: Micromachines- 2162873

Title: Application and Prospect of Artificial Intelligence Methods in Signal Integrity Prediction and Optimization of Microsystems

Journal: Micromachines

Dear reviewers and editor,

Thanks for the suggestive advice from reviewers and editor. The manuscript has been carefully checked again, and the English language has been improved. The followings are the response to the comments from the reviewers and editor. The corresponding modifications have been made and the important corrections are highlighted in YELLOW in the revised manuscript.

Reviewer #1:

  1. There is a typo on Page 2, Line 63 in the manuscript, “conclution”.

Response: Thanks for the reviewer’s suggestive comments. The misspelling has been corrected, and the manuscript has been carefully checked again, and the English language has been improved. The corresponding modifications have been made in ‘1 Introduction’. Thanks!

  1. The authors addressed an important yet interesting problem. From the manufacturing viewpoint, the signal integrity is the de facto critical bottleneck of DDR5 data lines/buses beyond 4800 mega-transfers per second. From the memory signal integrity perspective, it would be better to add an in-depth detailed discussion on the cutting-edge on-DIMM DDR5/DDR4 bus signal integrity and the eye diagram (on the prediction of eye height and eye width) of low-latency and high-speed DRAM microsystems, both at the DRAM chip level and the DIMM circuit board level. Example vendors include Changxin Memory Technologies, Micron, SK Hynix, and Samsung. The readers will be more interested with the manuscript. Theoretical analysis is expected to appear in the manuscript for publishing in Micromachines.

Response: Thanks for the reviewer’s suggestive comments. The current description of the microsystem signal integrity problem is defined. According to suggestions of reviewers, relevant literatures published by the above typical DDR manufacturers have been investigated in recent years. A related paper was found, Park [1] proposed a 192-Gb 896-GB/s 12-high stacked third-generation high-bandwidth memory, and proposed a layout technology based on deep learning to minimize signal delay deviation, and the proposed method improves the maximum read operation time margin by 33%. The corresponding modifications have been made in ‘1 Introduction’. Thanks!

[1] M. -J. Park et al., "A 192-Gb 12-High 896-GB/s HBM3 DRAM With a TSV Auto-Calibration Scheme and Machine-Learning-Based Layout Optimization," in IEEE Journal of Solid-State Circuits, vol. 58, no. 1, pp. 256-269, Jan. 2023, doi: 10.1109/JSSC.2022.3193354.

  1. The eye diagram is too small. It would be better to draw the eye diagram in Figure 3 larger to make the manuscript more intuitive

Response: Thanks for the reviewer’s suggestive comments. All images have been reconstructed in high definition from published publications. In particular, the eye image in Figure 3 has been enlarged. The corresponding modifications have been made in ‘2. Fast Prediction of Microsystem Performance by NNs’. Thanks!

  1. Why the authors did not consider the parasitic capacitance of transmission lines in their latency and power model?

Response: Thanks for the reviewer’s suggestive comments. In [36], the ANN method was used to establish a prediction model between design parameters and microstrip line parasitic parameter RLGC matrix, and then the predicted RLGC parasitic parameters were used to further solve differential impedance, common-mode impedance, near-end crosstalk, and far-end crosstalk performance parameters. Finally, the performance parameters are additionally optimized using a genetic algorithm and the corresponding design parameters for optimal performance are solved. The corresponding modifications have been made in ‘2. Fast Prediction of Microsystem Performance by Neural Networks’. Thanks!

[36] Kim, H.; Sui, C.; Cai, K.; Sen, B.; Fan, J. Fast and Precise High-Speed Channel Modeling and Optimization Technique Based on Machine Learning. IEEE Transactions on Electromagnetic Compatibility, 60, 2049–2052. Conference Name: IEEE Transactions on Electromagnetic Compatibility, https://doi.org/10.1109/TEMC.2017.2782704.

  1. The manuscript is full of abbreviations and the readability is poor. Please reduce that and use complete spelling, both in the figures and in the main text.

Response: Thanks for the reviewer’s suggestive comments. Some recurring terms in the article are abbreviated, the words themselves are lengthy, and the repetition rate in the article is strong. Some of the less commonly used acronyms were rewritten as their full names, while the more commonly used acronyms were clearly defined and marked with their abbreviations when they first appeared. At the same time, a list of acronyms is detailed at the end of the article. The corresponding modifications have been made in ‘Abbreviations’. Thanks!

  1. It would be better to summarize the training algorithm of the high-dimensional advanced neural networks. It seems that the authors only reviewed general methods. The domain-specific optimization of neural networks is not clear in the manuscript.

Response: Thanks for the reviewer’s suggestive comments. The domain-specific optimization methods of neural networks are investigated, In the signal integrity design of Microsystems, although the general neural network method can establish the mapping relationship between design parameters and performance parameters of Microsystems, it does not take into account the inherent physical characteristics and electromagnetic knowledge, which leads to the need for a large amount of data for neural networks and reduces the modeling efficiency of neural networks. In order to solve the above problems, Chen et al. [43] proposed a knowledge-based neural network method to design microwave devices, trained the nine design parameters of microstrip filter and its S parameters, used prior knowledge as the hidden layer of knowledge neurons, and then trained the neural network through particle swarm optimization algorithm. The microwave filter is designed on this basis. Na et al. [44] proposed an adaptive algorithm for automatic model structure for knowledge-based parametric modeling. L1-norm optimization is used to automatically determine the mapping in the knowledge-based model. The proposed method is used to design band-stop filters and to reduce the modeling time. Zhang et al. [45] proposed a method combining neural network and model order reduction, which solved the problem of pole/zero mismatch in the modeling of microwave passive devices by neural network and improved the modeling accuracy. The proposed method was applied to filter design with an average test error of only 1.37%. The corresponding modifications have been made in ‘2.1 Artificial Neural Network’.

Jin et al. [48] proposed a novel deep neural network structure for microwave components, which takes geometric parameters as input of the multi-layer hiding layer and frequency parameters as input of the first part of the hiding layer. The proposed structure can reduce the number of training parameters in deep neural network models and predict the performance of filters through the proposed structure. The maximum number of training parameters is reduced from 1224 to 574, which considerably reduces the training cost. The corresponding modifications have been made in ‘2.2 Deep Neural Network’.

Although it is possible to improve the performance of NN compared to traditional methods, from design parameters to the speed of mapping between them, direct use of classical neural network methods still incurs a large training cost. By conducting an in-depth analysis of SI problem to be solved, the neural network performance prediction method can be built with relevant knowledge to reduce the training cost and improve the extrapolation ability of the Neural Network. The corresponding modifications have been made in ‘2.5. Summary’. Thanks!

[43] Chen, Y.; Tian, Y.; Le, M. Modeling and optimization of microwave filter by ADS-based KBNN. International Journal of RF and Microwave Computer-Aided Engineering 2017, 27, e21062. doi:10.1002/mmce.21062.

[44] Na, W.; Feng, F.; Zhang, C.; Zhang, Q.J. A Unified Automated Parametric Modeling Algorithm Using Knowledge- Based Neural Network and l1 Optimization. IEEE Transactions on Microwave Theory and Techniques 2017, 65, 729–745. doi:10.1109/TMTT.2016.2630059.

[45] Zhang, J.; Chen, J.; Guo, Q.; Liu, W.; Feng, F.; Zhang, Q.J. Parameterized Modeling Incorporating MOR-Based Rational Transfer Functions With Neural Networks for Microwave Components. IEEE Microwave and Wireless Components Letters 2022, 32, 379–382. doi:10.1109/LMWC.2022.3146376.

[48] Jin, J.; Feng, F.; Zhang, J.; Yan, S.; Na, W.; Zhang, Q. A Novel Deep Neural Network Topology for Parametric Modeling of Passive Microwave Components. IEEE Access 2020, 8, 82273–82285. doi:10.1109/ACCESS.2020.2991890.

  1. Multiple typos occur in Table 1: “Predicte”.

Response: Thanks for the reviewer’s suggestive comments. The misspelling has been corrected, and the manuscript has been carefully checked again, and the English language has been improved. The corresponding modifications have been made in ‘Table 1’. Thanks!

  1. It would be better to organize the manuscript in terms of the optimization goals/performance metrics, i.e., the application fields in Table 1 the authors claimed.

Response: Thanks for the reviewer’s suggestive comments. The article structure suggested by the reviewer is also a reasonable idea to organize the paper. In this paper, The application of artificial intelligence methods to microsystem design is commonly divided into four steps \cite{2020Demystifying} : (1) clarify the problem to be solved, determine the design parameters and performance parameters; (2) obtain data; (3) establish a neural networks model, and use the acquired data to train neural networks to achieve performance prediction; (4) Optimize performance through intelligent optimization algorithm. Among them, performance prediction and performance optimization are two of the most crucial components of AI approaches in microsystem SI design, and in the performance prediction and performance optimization part from the perspective of methods to organize the article, and the structure of the paper is reasonable. Thanks.

  1. I guess that the “GA” in subsection 3.1 means the genetic algorithm. The authors should use complete spelling in their manuscript.

Response: Thanks for the reviewer’s suggestive comments. Full text is checked for acronyms and marked with full names where they first appear. The corresponding modifications have been made in ‘3.1 Genetic Algorithm’. Thanks!

  1. The insights derived from this review manuscript is relatively weak. The reviews on electrical-mechanical microsystems and sensor microsystems are missing.

Response: Thanks for the reviewer’s suggestive comments. The research status of electromechanical Microsystems and sensor Microsystems is investigated. Vijayara-ghavan et al. [10] proposed a high-performance computing microsystem based on 3D integration technology, integrating CPU, GPU and DRAM to achieve high throughput and efficient computing. Zaruba et al. [11] used advanced packaging technology to integrate computing cores with high-bandwidth memory into a high-performance memory microsystem with 25% lower power consumption than NVIDA Volta. Burd et al. [12] designed a computing microsystem through advanced packaging technology, with bandwidth up to 256GB/s and energy efficiency of only 1.2pj/bit. Vivet [13] et al. designed a high-performance computing microsystem based on a variety of processes. The designed microsystem has a 96-core processor and a signal delay of less than 0.6 ns/mm. Fotouhi et al. [14] designed an RF receiving and transmitting microsystem based on three-dimensional integration technology, integrating coupler, transceiver, array waveguide grating router, etc., which improved the computing performance by 23% and reduced the power consumption by 30%. Based on 3D integration technology, Shulaker et al. [15] proposed a microsystem integrating storage, computing, and sensors to realize high-performance information processing. Tang et al. [16] designed a MEMS gravimeter to achieve extreme sensitivity and large dynamic range through suspension design and optical displacement transducer. Yan et al. [17] designed a large capacitance trimethylamine sensor with linear sensitivity in the test concentration range, and developed a prototype sensor based on Co3O4@ZnO. Han et al. [18] used a recurrent neural network approach for noise reduction of 3D axial gyroscopes. Gao et al. [19] designed a MEMS filter with highly robust loan expansion capability by matching the network to broaden and enhance the out-of-band suppression, and applied an aluminum nitride S0 Lamb wave resonator into the filter to improve the loan expansion capability. The corresponding modifications have been made in ‘1 Introduction’. Thanks!

[10] Vijayaraghavan, T.; Eckert, Y.; Loh, G.H.; Schulte, M.J.; Ignatowski, M.; Beckmann, B.M.; Brantley, W.C.; Greathouse, J.L.; Huang, W.; Karunanithi, A.; et al. Design and Analysis of an APU for Exascale Computing. In Proceedings of the 2017 IEEE International Symposium on High Performance Computer Architecture (HPCA), 2017, pp. 85–96. doi:10.1109/HPCA.2017.42.

[11] Zaruba, F.; Schuiki, F.; Benini, L. A 4096-core RISC-V Chiplet Architecture for Ultra-efficient Floating-point Computing. In Proceedings of the 2020 IEEE Hot Chips 32 Symposium (HCS); IEEE Computer Society: Los Alamitos, CA, USA, 2020; pp. 1–24.

[12] Burd, T.; Beck, N.; White, S.; Paraschou, M.; Kalyanasundharam, N.; Donley, G.; Smith, A.; Hewitt, L.; Naffziger, S. “Zeppelin”: An SoC for Multichip Architectures. IEEE Journal of Solid-State Circuits 2019, 54, 133–143. doi:10.1109/JSSC.2018.2873584.

[13] Vivet, P.; Guthmuller, E.; Thonnart, Y.; Pillonnet, G.; Fuguet, C.; Miro-Panades, I.; Moritz, G.; Durupt, J.; Bernard, C.; Varreau, D.; et al. IntAct: A 96-Core Processor With Six Chiplets 3D-Stacked on an Active Interposer With Distributed Interconnects andIntegrated Power Management. IEEE Journal of Solid-State Circuits 2021, 56, 79–97. doi:10.1109/JSSC.2020.3036341.

[14] Fotouhi, P.; Werner, S.; Lowe-Power, J.; Yoo, S.J.B. Enabling scalable chiplet-based uniform memory architectures with silicon photonics. Proceedings of the International Symposium on Memory Systems 2019.

[15] Shulaker, M.M.; Hills, G.; Park, R.S.; Howe, R.T.; Saraswat, K.; Wong, H.S.P.; Mitra, S. Three-dimensional integration of nanotechnologies for computing and data storage on a single chip. 547, 74–78. doi:10.1038/nature22994, https://doi.org/10.1038/nature22994.

[16] Fotouhi, P.; Werner, S.; Lowe-Power, J.; Yoo, S.J.B. Enabling scalable chiplet-based uniform memory architectures with silicon photonics. Proceedings of the International Symposium on Memory Systems 2019.

[17] Shulaker, M.M.; Hills, G.; Park, R.S.; Howe, R.T.; Saraswat, K.; Wong, H.S.P.; Mitra, S. Three-dimensional integration of nanotechnologies for computing and data storage on a single chip. 547, 74–78. doi:10.1038/nature22994, https://doi.org/10.1038/nature22994.

[18] Tang, S.; Liu, H.; Yan, S.; Xu, X.; Wu, W.; Fan, J.; Liu, J.; Hu, C.; Tu, L. A high-sensitivity MEMS gravimeter with a large dynamic range. 5, 45. https://doi.org/10.1038/s41378-019-0089-7.

[19] Yan, W.; Xu, H.; Ling, M.; Zhou, S.; Qiu, T.; Deng, Y.; Zhao, Z.; Zhang, E. MOF-Derived Porous Hollow Co3O4@ZnO Cages for High-Performance MEMS Trimethylamine Sensors. ACS Sensors 2021, 6, 2613–2621, [https://doi.org/10.1021/acssensors.1c00315].

  1. The references section only has 50+ citations and is not adequate enough. Otherwise, the significance of this review manuscript is not up to the mark. Typical review papers are expected 200+ references for an essential readership.

Response: Thanks for the reviewer’s suggestive comments. Due to the relatively new research work on the application of artificial intelligence methods to microsystem signal integrity, the number of published high-quality research works is relatively small. The relevant literature has been fully investigated, and the number of references has increased to more than 100. The corresponding modifications have been made in ‘Reference’. Thanks!

Reviewer 2 Report

1.       Expand SI during first usage

2.       The  figure 3 (a) to (d) consisting of different results from different article. Do the authors have taken permission for reusing them.

3.       Similarly, 4, 5, 7 figures also have many results summarized. And some figure details are not visible clearly. Produce details having details which are readable ( with permission form the published Journals)

4.       What are the selecting criteria adopted for choosing papers and what is the span of duration considered for selecting papers.

Author Response

Ms. Ref. No.: Micromachines- 2162873

Title: Application and Prospect of Artificial Intelligence Methods in Signal Integrity Prediction and Optimization of Microsystems

Journal: Micromachines

Dear reviewers and editor,

Thanks for the suggestive advice from reviewers and editor. The manuscript has been carefully checked again, and the English language has been improved. The followings are the response to the comments from the reviewers and editor. The corresponding modifications have been made and the important corrections are highlighted in YELLOW in the revised manuscript.

Reviewer #2:

1.Expand SI during first usage

Response: Thanks for the reviewer’s suggestive comments. Full text is checked for acronyms and marked with full names where they first appear. The corresponding modifications have been made in ‘1 Introduction’. Thanks!

2.The figure 3 (a) to (d) consisting of different results from different article. Do the authors have taken permission for reusing them.

Response: Thanks for the reviewer’s suggestive comments. The pictures were used have taken permission. The corresponding modifications have been made in ‘2. Fast Prediction of Microsystem Performance by Neural Networks’. Thanks!

  1. Similarly, 4, 5, 7 figures also have many results summarized. And some figure details are not visible clearly. Produce details having details which are readable (with permission form the published Journals)

Response: Thanks for the reviewer’s suggestive comments. In this paper, some unnecessary results pictures are reduced. All images used have been applied for HD edition from the original publication journal(with permission form the published Journals). The corresponding modifications have been made in ‘2. Fast Prediction of Microsystem Performance by Neural Networks’ and ‘3. Intelligent Optimization Method for Microsystem Design’. Thanks!

4.What are the selecting criteria adopted for choosing papers and what is the span of duration considered for selecting papers.

Response: Thanks for the reviewer’s suggestive comments. Articles that use neural networks or intelligent algorithms to predict and optimize the performance of interconnect structures such as high-speed channels, microstrip lines, and microwave devices such as filters and couplers are selected by microsystems. The articles are mainly selected from influential international journals published in the last five years. For example, IEEE Transactions on Microwave Theory and Techniques, IEEE Transactions on Components, Packaging and Manufacturing Technology, IEEE Transactions on Electromagnetic Compatibility, IEEE Transactions on Computer-Aided Design for Integrated Circuits and Circuit Systems, and additional journals to ensure the authority of selected articles. Most of the selected articles were published within the last five years, ensuring the novelty of the work studied. Thanks!

Reviewer 3 Report

The authors propose the application of AI technology in microsystem SI performance prediction and optimization design, summarizes and compares the characteristics of the main neural networks methods of performance prediction and their application scenarios in microsystem SI design. Then summarizes and compares the characteristics of optimization design methods and application scenarios in microsystem SI optimization design. Finally, different prediction Algorithms and optimization Algorithms are discussed and compared.

The paper is interesting and is well writing. However, some major changes are needed:

- I have found two twice the expression “we” (first person of plural). Please use passive voice or third person. Check the document.

- In the last paragraphs of the introduction the contribution is commented but it is not clear. Please separate it from the rest of the paragraph and explain it further.

The fonts into the Fig. 3 f), g) and h), Fig 4. g), i) and j); and Fig. 5 c) and i) are very small. Please, improve this.

- The figure captions of Fig. 5 and 7 are very long. Please, reduce this and insert the description in the document text.

- The review of the state of the art is very focused on microsystems. However, it is important to talk about strategies such as those discussed in the article, first, in micro complex processes in general, i.e., micromanufacturing processes. To help in this, authors are recommended to review the following articles.

Beruvides, G., Quiza, R., Rivas, M. et al. Online detection of run out in microdrilling of tungsten and titanium alloys. Int J Adv Manuf Technol 74, 1567–1575 (2014). https://doi.org/10.1007/s00170-014-6091-1

Castaño F., Haber R.E., Mohammed W.M., Nejman M., Villalonga A., Martinez Lastra J.L.; Quality monitoring of complex manufacturing systems on the basis of model driven approach (2020) Smart Structures and Systems, 26 (4), pp. 495 - 506, DOI: 10.12989/sss.2020.26.4.495

Author Response

Ms. Ref. No.: Micromachines- 2162873

Title: Application and Prospect of Artificial Intelligence Methods in Signal Integrity Prediction and Optimization of Microsystems

Journal: Micromachines

Dear reviewers and editor,

Thanks for the suggestive advice from reviewers and editor. The manuscript has been carefully checked again, and the English language has been improved. The followings are the response to the comments from the reviewers and editor. The corresponding modifications have been made and the important corrections are highlighted in YELLOW in the revised manuscript.

Reviewer #3:

1.I have found two twice the expression “we” (first person of plural). Please use passive voice or third person. Check the document.

Response: Thanks for the reviewer’s suggestive comments. The whole text was checked and first person of plural were changed to the passive voice. The corresponding modifications have been made in ‘1 Introduction’ and ‘3. Intelligent Optimization Method for Microsystem Design’. Thanks!

2.In the last paragraphs of the introduction the contribution is commented but it is not clear. Please separate it from the rest of the paragraph and explain it further.

Response: Thanks for the reviewer’s suggestive comments. Contributions to the articles are listed separately and expanded accordingly. The corresponding modifications have been made in ‘1 Introduction’ (line 55-67). Thanks!

3.The fonts into the Fig. 3 f), g) and h), Fig 4. g), i) and j); and Fig. 5 c) and i) are very small. Please, improve this.

Response: Thanks for the reviewer’s suggestive comments. The pictures were remade and the clarity improved. The corresponding modifications have been made in ‘2. Fast Prediction of Microsystem Performance by Neural Networks’ and ‘3. Intelligent Optimization Method for Microsystem Design’. Thanks!

4.The figure captions of Fig. 5 and 7 are very long. Please, reduce this and insert the description in the document text.

Response: Thanks for the reviewer’s suggestive comments. Fig. 5 and Fig. 7 have been reworked and the notes have been streamlined. A detailed description is added to each picture. The corresponding modifications have been made in ‘2. Fast Prediction of Microsystem Performance by Neural Networks’ and ‘3. Intelligent Optimization Method for Microsystem Design’. Thanks!

5.The review of the state of the art is very focused on microsystems. However, it is important to talk about strategies such as those discussed in the article, first, in micro complex processes in general, i.e., micromanufacturing processes. To help in this, authors are recommended to review the following articles.

Beruvides, G., Quiza, R., Rivas, M. et al. Online detection of run out in microdrilling of tungsten and titanium alloys. Int J Adv Manuf Technol 74, 1567–1575 (2014). https://doi.org/10.1007/s00170-014-6091-1

Castaño F., Haber R.E., Mohammed W.M., Nejman M., Villalonga A., Martinez Lastra J.L.; Quality monitoring of complex manufacturing systems on the basis of model driven approach (2020) Smart Structures and Systems, 26 (4), pp. 495 - 506, DOI: 10.12989/sss.2020.26.4.495

Response: Thanks for the reviewer’s suggestive comments. The article and book recommended by reviewer are highly relevant to this research, and they have been cited in this paper. The corresponding modifications have been made in ‘References’. Thanks!

Round 2

Reviewer 1 Report

The authors addressed the problems raised by the referees point-by-point in the main text of this conference-extended manuscript with a fine revision. The authors also added an interesting topic - model order reduction – for computationally efficient computer-aided linear system simulation composing both the eigen mechanisms and the machine learning data for solving challenging scientific and engineering problems. Having said that, as per the current version, the authors’ review to the literature is not insightful and comprehensive enough. I would suggest the manuscript could be conditionally accepted undergoing some mandatory modifications applied to the current review manuscript to improve the quality.

1. It seems that there is a fault appeared on Ref. [56] in the revised manuscript. I believe IEEE Transactions on Electromagnetic Compatibility should be a Journal Name, not a conference.

2. As the physics insights from this review manuscript is sub-par, I suggest that the manuscript would be benefit from summarizing the challenges and prospects of heterogeneous-integrated microsystems at multiscale for microengineering.

3. Except for the common MEMS microsystems, the manuscript would be a big plus from adding a review on the state-of-the-art MOEMS microsystems. Example references can be found at the Journal of Micromechanics and Microengineering (JMM) and Journal of Micro/Nanopatterning, Materials, and Metrology (JM3).

4. There are several “[?]” in Table 1 and Table 2.

Author Response

Ms. Ref. No.: Micromachines- 2162873

Title: Application and Prospect of Artificial Intelligence Methods in Signal Integrity Prediction and Optimization of Microsystems

Journal: Micromachines

Dear reviewers and editor,

Thanks for the suggestive advice from reviewers and editor. The manuscript has been carefully checked again, and the English language has been improved. The followings are the response to the comments from the reviewers and editor. The corresponding modifications have been made and the important corrections are highlighted in YELLOW in the revised manuscript.

Reviewer #1:

The authors addressed the problems raised by the referees point-by-point in the main text of this conference-extended manuscript with a fine revision. The authors also added an interesting topic - model order reduction – for computationally efficient computer-aided linear system simulation composing both the eigen mechanisms and the machine learning data for solving challenging scientific and engineering problems. Having said that, as per the current version, the authors’ review to the literature is not insightful and comprehensive enough. I would suggest the manuscript could be conditionally accepted undergoing some mandatory modifications applied to the current review manuscript to improve the quality.

  1. It seems that there is a fault appeared on Ref. [56] in the revised manuscript. I believe IEEE Transactions on Electromagnetic Compatibility should be a Journal Name, not a conference.

Response: Thanks for the reviewer’s suggestive comments. All the reference were checked and the Ref. [56] was corrected. The corresponding modifications have been made in ‘1 Introduction’. Thanks!

  1. As the physics insights from this review manuscript is sub-par, I suggest that the manuscript would be benefit from summarizing the challenges and prospects of heterogeneous-integrated microsystems at multiscale for microengineering.

Response: Thanks for the reviewer’s suggestive comments. Combined with the challenge of three-dimensional heterogeneous integration, the future application scenarios of artificial intelligence methods in microsystems are further supplemented.

The system is greatly reduced in size and integrated with multiple components. Although the system performance is greatly improved, the resulting signal crosstalk, cross-scale, multi-field coupling, and other issues make the signal integrity design more complex and time-consuming.

In the context of SI prediction in microsystems, neural networks are the main AI methods, which are mainly used in high-speed signal path-eye map prediction, crosstalk prediction, parasitic parameter prediction, frequency response prediction, etc. Using the obtained data to train a NN, the traditional electromagnetic/circuit simulation model is replaced by NN and greatly improve the efficiency of simulation. For different application scenarios, the NN structure suitable for the problem should be selected based on the characteristics of different NNs. In the future, the system volume will be further reduced, multi-field coupling effects will be more severe, and the trade-off relation between multiple software iterations and multiple performance metrics will be complicated, which will lead to a lower efficiency when using traditional analysis methods. In addition, the meshing and solution times will be further improved when the multi-scale components are integrated in Microsystems. AI methods may be an effective approach to solve the above problems. By solving for the weights of the hidden layers, the flow of data from design parameters to performance metrics during the multi-software iteration can be constructed to reduce the design difficulty. In addition, neural network models can be constructed to skip steps such as grid partitioning and time-consuming steps due to cross-scale effects, thus improving simulation efficiency.

The corresponding modifications have been made in ‘1 Introduction’ and ‘4. Discussions And Outlook’. Thanks!

  1. Except for the common MEMS microsystems, the manuscript would be a big plus from adding a review on the state-of-the-art MOEMS microsystems. Example references can be found at the Journal of Micromechanics and Microengineering (JMM) and Journal of Micro/Nanopatterning, Materials, and Metrology (JM3).

Response: Thanks for the reviewer’s suggestive comments. The literature on MOEMS Microsystems was researched. The corresponding modifications have been made in ‘1 Introduction’. Thanks!

Mohammadian et al. [23] designed an optical XOR logic gate based on a ring resonator and a Micro-electromechanical system (MOEMS), and established a finite element model of the optical ring resonator to improve the wavelength shift. Rochus et al. [24] proposed a nonlinear mechanical and optical loss of micro optical mechanical pressure sensor fast modeling method, considering the strong coupling nonlinear mechanics model, analyzed the location based on membrane size, residual stress, waveguide, optical wavelength and optical machine coupling effect on the phase rule. TAGHAVI et al. [25] proposed a kind of based on closed-loop accelerometer of MOEMS cloth interferometer in the method, the design of closed-loop MOEMS accelerometer has a wider measuring range and higher sensitivity.

[23] Mohammadian, S.; Babazadeh, F.; Abedi, K. Study of a MOEMS XOR gate based on optical ring resonator. Physica Scripta 2021,96, 125532. doi:10.1088/1402-4896/ac3ea2.

[24] Rochus, V.; Jansen, R.; Goyvaerts, J.; Neutens, P.; O’Callaghan, J.; Rottenberg, X. Fast analytical model of MZI micro-opto-mechanical pressure sensor. Journal of Micromechanics and Microengineering 2018, 28, 064003. doi:10.1088/1361-6439/aab461.

[25] Taghavi, M.; Abedi, A.; Parsanasab, G.M.; Rahimi, M.; Noori, M.; Nourolahi, H.; Latifi, H. Closed-loop MOEMS accelerometer. Opt. Express 2022, 30, 20159–20174. doi:10.1364/OE.455772.

  1. There are several “[?]” in Table 1 and Table 2.

Response: Thanks for the reviewer’s suggestive comments. All references and citations in the paper are checked, and incorrect citations in the text are corrected. The corresponding modifications have been made in ‘1 Introduction’. Thanks!

Reviewer 3 Report

The authors have modified the paper according to the reviewer´s comments. For this, the article have improved in terms of scientifi soundness and quality of presentation.

Author Response

Thanks for the reviewer’s suggestive comments for  the paper.

Round 3

Reviewer 1 Report

The authors have addressed all my concerns point-by-point in the manuscript. The quality of this manuscript is significantly improved by the revision. The manuscript is of great interest and is recommended for publication.